# Trends and Patterns of ICU E-Referrals in Saudi Arabia during 2020–2021: Results from the National Saudi Medical Appointments and Referrals Centre

**DOI:** 10.3390/healthcare12191939

**Published:** 2024-09-27

**Authors:** Abdullah A. Alharbi, Nawfal A. Aljerian, Hani A. Alghamdi, Meshary S. Binhotan, Ali K. Alsultan, Mohammed S. Arafat, Abdulrahman Aldhabib, Ahmed I. Aloqayli, Eid B. Alwahbi, Mohammed A. Muaddi, Mohammed K. Alabdulaali

**Affiliations:** 1Family and Community Medicine Department, Faculty of Medicine, Jazan University, Jazan City 45142, Saudi Arabia; mothman@jazanu.edu.sa; 2Medical Referrals Centre, Ministry of Health, Riyadh 12382, Saudi Arabia; njerian@moh.gov.sa (N.A.A.); alkalsultan@moh.gov.sa (A.K.A.); arafatms@moh.gov.sa (M.S.A.); aaldhabib@moh.gov.sa (A.A.); aialoqayli@moh.gov.sa (A.I.A.); ealwahbi@moh.gov.sa (E.B.A.); 3Emergency Medicine Department, King Saud bin Abdulaziz University for Health Sciences, Riyadh 11481, Saudi Arabia; 4Department of Family and Community Medicine, College of Medicine, King Saud University, Riyadh 11461, Saudi Arabia; halhajalah@ksu.edu.sa; 5Emergency Medical Services Department, College of Applied Medical Sciences, King Saud bin Abdulaziz University for Health Sciences, Riyadh 11426, Saudi Arabia; hotanm@ksau-hs.edu.sa; 6King Abdullah International Medical Research Centre, Riyadh 11481, Saudi Arabia; 7Ministry of Health, Riyadh 12382, Saudi Arabia; mal-abdul-aali@moh.gov.sa

**Keywords:** epidemiology, ICU, COVID-19, pandemic response, healthcare system, SMARC, Saudi Arabia

## Abstract

**Background/Objectives:** Intensive care unit (ICU) e-referrals are an important indicator for exploring potential deficiencies in critical care resources. This study aimed to examine the epidemiology and patterns of ICU e-referrals across all regions of Saudi Arabia during the COVID-19 pandemic using routinely collected data from April 2020 to December 2021. **Methods:** This descriptive epidemiological study analyzed data from the Saudi Medical Appointments and Referrals Centre (SMARC). This study reveals novel regional ICU e-referral patterns for critical cases using national unique digital health data, adding insights beyond the existing literature. Variables included age, sex, referral timing, reason, specialty, and region of origin. Descriptive statistics and mapping of administrative areas were performed based on e-referral request rates per 10,000 population. **Results:** During the study period, 36,619 patients had ICU e-referral requests. The mean age was 54.28 years, with males constituting 64.81% of requests. Out-of-scope e-referrals comprised 71.44% of requests. Referrals related to medical specialties, such as cardiology and pulmonology, were the most common (62.48%). Referral patterns showed peaks in July–August 2020 and May 2021. The Northern Border and Albaha areas had the highest request rates per population, potentially reflecting a higher proportion of severe cases requiring ICU-level care compared to other regions. **Conclusions:** The temporal pattern and geographic distribution of ICU e-referrals mirrored previously reported critical COVID-19 cases in Saudi Arabia. Preventive measures and vaccination programs contributed to a significant decline in ICU e-referral requests, suggesting a positive impact on controlling severe COVID-19 cases. Population-adjusted analysis revealed regional disparities, highlighting the importance of considering population size in healthcare resource management and policy.

## 1. Introduction

The healthcare system in the KSA currently encompasses 13 administrative areas, though, under Vision 2030, these are consolidated into five larger business units (BUs) (Central, Northern, Southern, Eastern, and Western) to provide services to the population of nearly 34 million through integrated public and private facilities. The system is being upgraded to align with the objectives of the 2030 Vision and Sustainable Development Goals [1,2]. An optimal e-referral system is an essential component of high-quality health care, utilizing an efficient structure and promoting successful interaction across all levels of care, providing referred patients with the highest quality of care [3]. Multiple factors affect the productivity of an e-referral system, including the healthcare system as a whole, communication infrastructures, human and non-human resources, technological advances, and patient reactions to e-referrals [4].

The e-referral system in Saudi Arabia, which was formerly known as Ehalati and recently renamed the Saudi Medical Appointments and Referrals Centre (SMARC), was established in 2012 and expanded in 2018 to integrate data between public and private health facilities across the country [5]. The SMARC system provides a centralized platform to efficiently process large volumes of e-referrals across all regions of Saudi Arabia. During the COVID-19 pandemic, SMARC proved uniquely capable of tracking ICU bed availability and e-referral requests in real time across the healthcare system. Key features like its integrated structure and robust data analytics capabilities enabled insights into the impact of COVID-19 on critical care capacity nationally and by region. By leveraging the detailed ICU e-referral data collected by SMARC throughout the pandemic, this study can provide valuable analysis of trends and disparities in a way not possible without this system.

On 2 March 2020, the first case of Coronavirus Disease 2019 (COVID-19) was reported in the Kingdom of Saudi Arabia (KSA). This signaled the beginning of the COVID-19 era, leading to a nationwide lockdown, which subsequently led to profound economic and social disruptions within the country [1]. The pandemic also had significant impacts on routine healthcare and treatment pathways across all medical specialties. This impact was particularly profound in intensive care medicine, which was focal in its response to the pandemic. The rapid rate of transmission, as well as the increasing number of intensive care unit (ICU) admissions among hospitalized patients, demonstrated significant challenges to healthcare systems globally [6,7,8].

Early in the pandemic, approximately half of all patients hospitalized with COVID-19 in the KSA were admitted to the ICU, compared to only one-third in Wuhan, China [1,2,9]. This variation is most likely due to the varying levels of severity, as well as a range of risk factors such as advanced age, presence of comorbidities, and behavioral factors [8,10]. In response to the outbreak, Saudi Arabia implemented various preventive measures such as curfews and social distancing aimed at reducing further severe cases requiring ICU-level care. While these interventions likely influenced healthcare utilization and ICU e-referral patterns to some degree, this study focuses specifically on analyzing overall national and regional trends and patterns in ICU e-referrals rather than isolating the impact of preventive measures. Their variable effects across different regions were outside the scope of this descriptive analysis. Healthcare systems also took measures to slow the potential of transmission, such as changes to elective surgeries and inpatient access. These were restructuring efforts aimed at managing the rising number of cases. In addition, temporary facilities were established for the purpose of managing care specifically for COVID-19 patients [11].

The pandemic caused an overwhelming demand for critical care resources, straining healthcare systems and ICUs worldwide and requiring a rapid expansion of critical care capacity. An asymmetry between supply and demand of ICU capacity has significant negative impacts on patient outcomes as well as on the well-being of the healthcare team [12,13]. In order to meet the demand for increased critical care, hospitals expanded ICU bed capacity. They created surge beds outside recognized ICUs, leading to staffing and resource challenges, such as reduced staffing ratios and redeployment of inexperienced personnel. Equipment shortages, including mechanical ventilators and medications, were also reported [14,15]. Clearly, the pandemic caused a significant disruption in routine care while also emphasizing the importance of ICU capacity in preparedness planning [16].

While numerous studies have examined ICU capacities and pandemic responses globally, limited data exist on referral patterns, representing a knowledge gap this study aims to address. By leveraging national digital health data from the centralized SMARC referral system, this study uniquely analyses ICU referral trends across Saudi Arabia’s 13 regions during the critical COVID-19 pandemic period. Most of the literature focuses narrowly on data from a limited number of hospitals or local intra-hospital ICU referrals and capacity management [17,18,19]. In contrast, our comprehensive national-level analysis of both intra-regional and inter-regional transfers provides novel insights into broader ICU referral dynamics, addressing a significant gap in current research concentrated on isolated hospitals. Examining these referral patterns during the pandemic offers vital perspectives into healthcare system performance and resource allocation.

## 2. Materials and Methods

### 2.1. Setting and Data Source

The data used to reach the objectives of this research were obtained from the SMARC via the Unified System of Medical Referrals (USMR). In the KSA, and as part of the healthcare system, the USMR may be accessed by public hospitals within the MoH’s network as well as by private hospitals. This is performed through an appointment and medical e-referral office within those hospitals. A typical scenario involves this department receiving an ICU e-referral request from the treating physician, which is then uploaded to the USMR. The uploaded requests usually indicate whether the e-referral is prioritized by designating it as an emergency vs. not an emergency and recommend up to three potential hospitals that offer the requested services. If no hospitals are chosen, the USMT automatically chooses three potential hospitals with the required resources located within the same region. The SMARC system has a time limit of 72 h for emergency e-referrals and 14 days for routine non-emergency e-referrals. If the time is lapsed with no decision, the e-referral request is then escalated to the SMARC medical e-referral management to find an appropriate alternative option from any of the available public or private hospitals either within the same region or externally.

In addition, the SMARC system also provides a 24-h lifesaving hotline service known as “1937”. This service is available to immediately accept e-referral requests for lifesaving critically ill patients. Through this service, the treating physician calls this hotline and requests an ICU e-referral. This call is received by a lifesaving agent, who then transfers the call to an on-call ICU consultant for review and acceptance. In the usual case of immediate acceptance, the treating physician then uploads the requests to the USMR with the acceptance code received from the 1937 lifesaving hotline service. However, in case the e-referral is not accepted, the ICU consultant then guides the treating physician to upload an emergency e-referral through the USMR. Figure 1 describes the ICU e-referral process.

### 2.2. Study Design

This is a descriptive retrospective analysis of national data on all ICU e-referrals collected through the routine use of the SMARC e-referral system. The dataset includes the full census of ICU e-referrals submitted via SMARC from April 2020 to December 2021.

### 2.3. Ethical Considerations

This study adheres to the ethical guidelines of the World Medical Association Declaration of Helsinki [20]. Ethical approval was acquired from the Ministry of Health central institutional review board (IRB log No: 23-77-E on 20 September 2023). All precautions in terms of anonymity, patient confidentiality, and privacy of patients’ data were taken.

### 2.4. Measurements

This research utilized all available variables collected by the SMARC system. These variables included sociodemographic characteristics such as the age, sex, and nationality (Saudi vs. non-Saudi) of patients for which the ICU e-referral was requested, as well as the region of the request and the BUs from which the request originated, and the month and year of the e-referral request. Referral specific characteristics were also used; these included the reason for e-referrals (out of scope, i.e., an unavailable specialty of specialized physician, overcapacity and bed shortage, unavailable equipment, unavailable bed or compassionate request) and the specialty for which the request belonged (surgical, cardiac surgery, obstetrics, gynecology, oncology, or organ transplantation).

### 2.5. Statistical Analysis

Cross-tabulations in accordance with the five BUs were performed with each of the variables, and tests of significance were obtained through Chi-squared and ANOVA tests. This enabled us to present the epidemiological patterns of e-referrals between the different BUs and 13 administrative areas. To further clarify these patterns, color-coded maps were drawn in ArcGIS software (ArcGIS 10.0, Environmental Systems Research Institute, Inc., Redlands, CA, USA) according to the percentage of e-referrals in each administrative area.

## 3. Results

### 3.1. Sociodemographic Characteristics of ICU Patients with E-Referral Requests

The total number of patients with an ICU e-referral request from the KSA during 2020 and 2021 was 36,619. The overall mean age was 54.28 ± 20.46 years. Males were found to have more requests than females (64.50% and 35.50%, respectively). Around 71% of all requests were for Saudi nationals, and the year 2021 had more requests compared to its predecessor (Table 1).

### 3.2. Sociodemographic Characteristics of ICU Patients with E-Referral Requests According to BUs

Table 2 shows the sociodemographic characteristics of patients with an ICU e-referral request according to the five BUs. Patients with requests from the Southern BUs were slightly older than those from other BUs. The Western BUs had the highest proportion of male requests, whereas the Eastern BUs had the lowest proportion (66.29% and 60.03%, respectively). Non-Saudis constituted over a quarter of all ICU e-referral requests from the Central BUs, and this was the highest proportion for non-Saudis across all BUs. Across all BUs, the e-referral requests were higher in 2021 compared to 2020. All sociodemographic differences and e-referral patterns across BUs noted here were statistically significant at *p* < 0.001, as detailed in Table 2.

### 3.3. Referral Characteristics of ICU Patients

Table 3 presents the results of ICU e-referral requests according to the five BUs. Out of the five reasons for ICU e-referrals, the most common reason was for the case to be out of scope. The least reported reason was social (0.60%). The BU that mostly reported the reason for e-referral as out of scope was the Central BU (84.60%), and the Western BU was the least to report it at only 64.38%. Overcapacity and bed shortage were mostly reported in Western BU (30.09%) and the least reported in Central BU (09.48%). The unavailability of a machine was reported more often in the Northern and Southern BUs (9.73% and 9.32%, respectively).

As for specialties, the most commonly reported specialty for ICU e-referrals was medical (62.48%). Medical specialties were the most common specialty of e-referrals from the Western BU and the least common from the Central BU, whereas surgical specialties, as well as cardiac surgery, mostly originated from the Central BU. The Northern BU mostly requested e-referrals for oncology. All differences in e-referral reasons and specialties across BUs highlighted here were statistically significant at *p* < 0.001, as shown in Table 3.

### 3.4. Epidemiological Trends and Patterns of ICU E-Referral Requests

Figure 2 shows the trends of ICU e-referral requests across months separately for 2020 and 2021. Since April 2020, the proportion of ICU requests has been minimal, peaking in July and August, whereas for 2021, the year commenced with low proportions of requests, slightly increasing till it reached the peak in May 2021 and gradually decreasing toward the end of the year.

Figure 3 presents the pattern of quintiles of ICU e-referral requests per 10,000 of the population for the 13 administrative areas of the KSA. Both the Northern Border and Albaha had the highest quintiles of ICU e-referral requests, whereas Riyadh, Eastern Province, and Tabuk had the lowest quintiles.

### 3.5. Rate of ICU E-Referrals According to Business Units and Administrative Areas

Table 4 and Figure 4 present the rate of ICU e-referrals per 10,000 of the population according to both BUs and administrative areas. With regards to BUs, although the Western BU had the highest number of requests, examining these requests per 10,000 of the populations shows that the Southern BU was the highest in terms of e-referrals followed by the Northern BU (16.64 and 15.10 per 10,000, respectively). Similarly, for the administrative areas, the Albaha administrative area had a high number of ICU e-referral requests per 10,000 of its population (49.41) despite the fact that the crude number of e-referrals was relatively small. A similar observation is seen for the Northern border administrative area (37.42 per 10,000).

## 4. Discussion

This study is the first to use secondary collected data by the SMARC system of the KSA to identify epidemiological patterns and trends of ICU patients’ e-referrals. This system is unique in that it is limited to secondary, tertiary, and specialized healthcare facilities with no primary care involvement. Hence, its ability to capture patterns of critical illness is heightened, especially since the data captured the period in which COVID-19 was highly active and severe cases were at a peak. The current analysis shows variations in the number of ICU requests, reasons for e-referrals, and clinical specialties involved across both the BUs and administrative areas.

During the pandemic, various operational adjustments were made to isolate COVID-19 patients when feasible. However, detailed data on implementation approaches were outside the scope of this study. The existing literature indicates that Saudi Arabia followed general recommendations to separate COVID-19 patients from non-infected patients [21,22]. Further research is needed to examine how referral patterns interacted with ICU organization strategies aimed at minimizing transmission risks. Investigating the impacts of isolating COVID units could offer insights to optimize future pandemic responses, balancing infection control with equitable access.

### 4.1. Variations in ICU E-Referral Characteristics in General

The average age of patients with an ICU referral request was found to be slightly younger than the mean age of ICU admitted patients in the KSA [23]. This difference between admitted cases and cases with e-referral requests may be due to their clinical nature, the severity of the disease, and the capacity of management. It may be that e-referrals are requested for cases where patients’ outcomes are anticipated to be improved. Differences were also seen for sex, where males had almost double the female requests. Considering that this period covers the peak of the pandemic and the fact that global trends show that males are susceptible to more severe consequences from COVID-19 than females, it has been reflected in this current analysis [24]. Furthermore, e-referrals for Saudis were almost triple that of non-Saudis, which is proportional to the Saudi population census. The Saudi government also announced on 30 March 2020 that non-Saudis, including immigrants and illegal residents, will have free and unconditional access to healthcare and treatment in public facilities without any financial or legal liabilities during the pandemic response period [25]. It is worth noting that non-Saudi workers also benefit from access to private healthcare due to insurance coverage by their employers [26].

According to the characteristics of ICU e-referrals, the “out of scope” category was predominant. ICU care requires a multidisciplinary team, including medical and surgical specialists, to diagnose, treat, and monitor different types of critical illnesses, and the shortage of ICU staff is a well-known issue that existed even before the pandemic and persists in advanced healthcare systems [27,28]. Naturally, this issue of shortage of staff was further exacerbated by the increased demand for ICU admissions due to critical cases of COVID-19 [29].

Similarly, resource constraints emerged as the second reason for ICU e-referral during the pandemic. Treating COVID-19 patients requires sufficient availability of hospital resources, including vacant ICU beds, personal protective equipment, and well-trained ICU staff [16]. The swift and high increase in cases overwhelmed these resources during the response period [30]. This shortage resulted in straining the public health resources and capacities of public hospitals worldwide [31].

To overcome the high surge in ICUs, the private sector had to assist in the response. In the KSA, a strategic approach was implemented whereby patients were referred to private hospitals, with the costs covered by the Saudi government, regardless of nationality or residential status. A further policy mandated the designation of specific healthcare facilities as reference centers and the development of a system to daily monitor the isolation bed occupancy. This facilitates timely decisions for bed expansions or patient transfers to neighboring facilities [32]. This goal is to effectively manage ICU capacity, ensuring sufficient availability during this and any future crisis. By leveraging the resources of private healthcare facilities, the government aims to maintain a buffer in ICU occupancy, thus enhancing the healthcare system’s preparedness for potential emergencies.

Further analysis of e-referral indications by specialty could provide more nuanced insights into these patterns.

During the pandemic, a variety of supplies are critical for providing lifesaving care to patients. The shortage of ventilators and medical staff is significant, but the scarcity of other resources like personal protective equipment, monitors, intravenous supplies, and medications also critically affects patient outcomes and limits the capacity for delivering effective critical care [33]. The Saudi pandemic response focused on equipment availability by forming a national committee for critical care equipment, tasked with determining the types and quantities of equipment needed for a worst-case scenario of 10,500 simultaneous critical COVID-19 cases [34]. This planning guided the acquisition of critical care hardware, consumables, and medications. This systematic approach ensured preparedness for escalating ICU demands.

### 4.2. Variations in ICU E-Referral Requests According to BUs and Regions

This study found that Eastern BUs had the highest proportion of ICU e-referral requests due to unavailable beds. According to the MoH statistical yearbook for 2022, the ICU had the least number of ICU beds compared to other regions [35]. However, this region ranks third in terms of population size [36]. Hence, our results show a disproportionate allocation of ICU beds specifically for this region. This study also showed that e-referrals due to unavailable equipment and machines were the highest in the Northern and Southern BUs. These disparities offer insights into the regional variations in healthcare resource allocation. In a previously published study that examined the rate of confirmed COVID-19 ICU admissions as a quality indicator between the five BUs, they found that the highest odds of admission were in the Northern and Western BUs [37]. The conclusion was that patients’ clinical characteristics and resource allocation were potential reasons for these variations. This study further emphasizes that healthcare resource management is of vital importance.

Adjusting for population size is critical in understanding the demand for services. For example, although the results show that the highest proportion of requests was observed in the Western BUs, it showed an e-referral rate of 14.88 per 10,000 of the population, which ranks third. However, when further investigating administrative areas shows that the Albaha area, which belongs to the Western BU, had an extremely high ICU e-referral request rate (49.41 per 10,000). The relatively smaller population residing in this area may indicate a higher severity of ICU cases or an extreme shortage of resources.

Medical specialty accounted for almost two-thirds of the specialties requested. This is not surprising since these data cover the period when COVID-19 began spreading in Saudi Arabia. Respiratory illnesses from COVID-19 drove the high demand for critical care management by medical specialties [38].

The Western BU had the highest proportion of e-referral requests. This aligns with previous reports showing the highest COVID-19 mortality in this region [1]. For oncology e-referrals, the highest proportion came from the Northern BU. According to the MoH, there are no oncology centers in the north. However, the MoH regulates four fully equipped oncology centers in Saudi Arabia, with two in the Central region and one each in the Eastern and Western regions [35]. The lack of local oncology infrastructure in the north likely contributed to the high e-referral rates, as patients depended on external ICUs for critical care.

This centralization approach is a known healthcare strategy to improve outcomes of patients experiencing specific critical conditions, including oncology, and is followed in the UK to allow for swift access to multidisciplinary teams and expert professionals [39,40]. Nevertheless, the absence of a local oncology center meant more e-referrals for the Northern BU. Similar disparities in regional resources may have driven higher e-referral rates in other areas as well. The Saudi Vision 2030 plan aims to address these healthcare access disparities through reforms to build capacity across all regions.

### 4.3. Patterns of ICU E-Referral Requests over Time

The pattern of monthly ICU e-referral requests during the two-year period showed two main waves where the requests were high. This pattern was significantly influenced by the spread and severity of COVID-19 cases across the country. The first wave of ICU e-referral requests reached its peak in July 2020 and ended by February 2021, coinciding with a peak in COVID-19 deaths and critical cases. This aligns with July 2020, seeing the highest number of daily deaths (58) and critical cases (2295) during the first surge [41].

These trends mirror global evidence that ICU admissions and e-referrals rise and fall in line with COVID-19 cases [38,42]. However, healthcare policies likely also played a role. It is worth noting that these patterns of incline and decline found in these data may also be affected by the national strategies put forth by the government to mitigate the surge in critical cases. In addition to the suspension of flights, the shift to online education, the suspension of social and religious gatherings, as well as quarantine, the introduction of the vaccination program has also played a vital role in this strategy [21,43]. The decline in the second ICU e-referral request wave in June 2021 mirrored the progress of the COVID-19 vaccination plan when 50% of the Saudi population (aged ≥12 years) received the first vaccination dose, and this decline was sharper toward August 2021, when 50% of the population had received the second dose. Vaccinations have proven effective in reducing severe infections and deaths [44], which would directly impact ICU e-referral needs. Further research could delineate the relative influence of COVID-19 epidemiology versus healthcare interventions on the ICU e-referral fluctuations.

### 4.4. Strengths and Limitations

This study provides a valuable initial analysis of ICU e-referral patterns during the COVID-19 pandemic in Saudi Arabia using a comprehensive national digital dataset. However, there are important limitations to consider. As a retrospective review of routine e-referral data, details were lacking on COVID-19 diagnoses, reasons for ICU admission, and types of ICUs needed. Data constraints also precluded the assessment of ICU capacity and healthcare workforce factors that underlined referral trends. Limited data on regional healthcare infrastructure restricted more in-depth analysis of drivers behind referral disparities between regions. While these limitations prevent advanced statistical modeling and conclusive identification of predictors, this study yields useful descriptive epidemiology of ICU e-referral trends nationally and regionally.

This study was further limited by the inability to track referral reasons and out-of-scope referrals longitudinally. Follow-up efforts with expanded data elements are warranted to enable a more rigorous analysis of healthcare utilization and outcomes. Prospective data collection, including clinical details and capacity factors, would facilitate future modeling of contributors to ICU e-referral patterns during the pandemic response.

Despite limitations, this initial investigation of routinely collected data provides valuable insights to inform ICU planning and pandemic preparedness. However, the findings require interpretation within the context of Saudi Arabia’s healthcare system. While there are transferable lessons, policies should be adapted to consider local needs and resources. Further research building on this groundwork with enriched datasets can substantively advance the understanding of optimizing critical care delivery during public health crises.

## 5. Conclusions

This nationwide analysis of ICU e-referrals during the first two years of the COVID-19 pandemic in Saudi Arabia elucidates utilization patterns to inform capacity planning. Findings mirror reported critical case trends, peaking early and then declining with public health measures. Adjusting for population size reveals unequal burdens, with certain regions requiring targeted ICU investment aligned with the Saudi Vision 2030 plan’s goals to build healthcare access across all areas. Medical specialties predominated, reflecting respiratory care demands. Centralization may increase e-referrals where local capacity is limited. Despite data limitations, these insights strengthen Saudi Arabia’s pandemic preparedness and critical care delivery for current and future crises. Follow-up research could enable predictive modeling to strategically optimize ICU readiness. By illuminating key epidemiologic patterns and informing strategic growth, this study makes an important contribution to building national ICU capacity at the frontlines of healthcare crises requiring lifesaving critical care.

## Figures and Tables

**Figure 1 healthcare-12-01939-f001:**
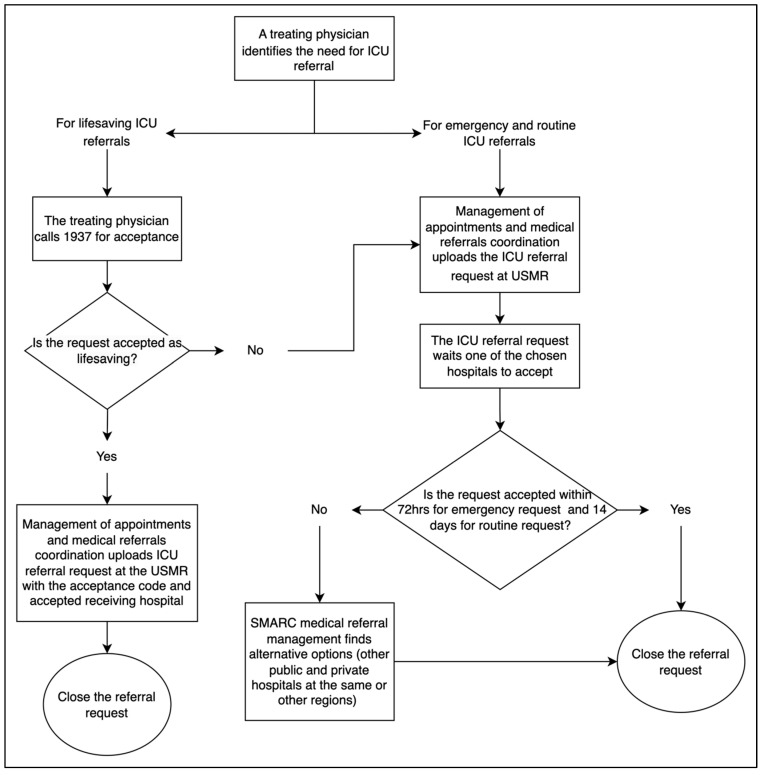
A typical ICU e-referral process.

**Figure 2 healthcare-12-01939-f002:**
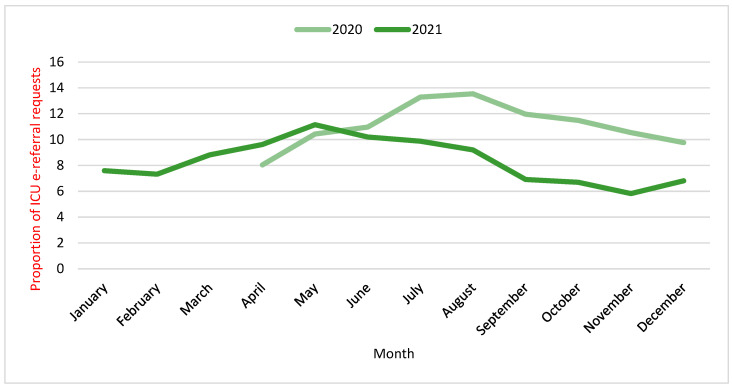
Trends of ICU e-referral requests for 2020 and 2021.

**Figure 3 healthcare-12-01939-f003:**
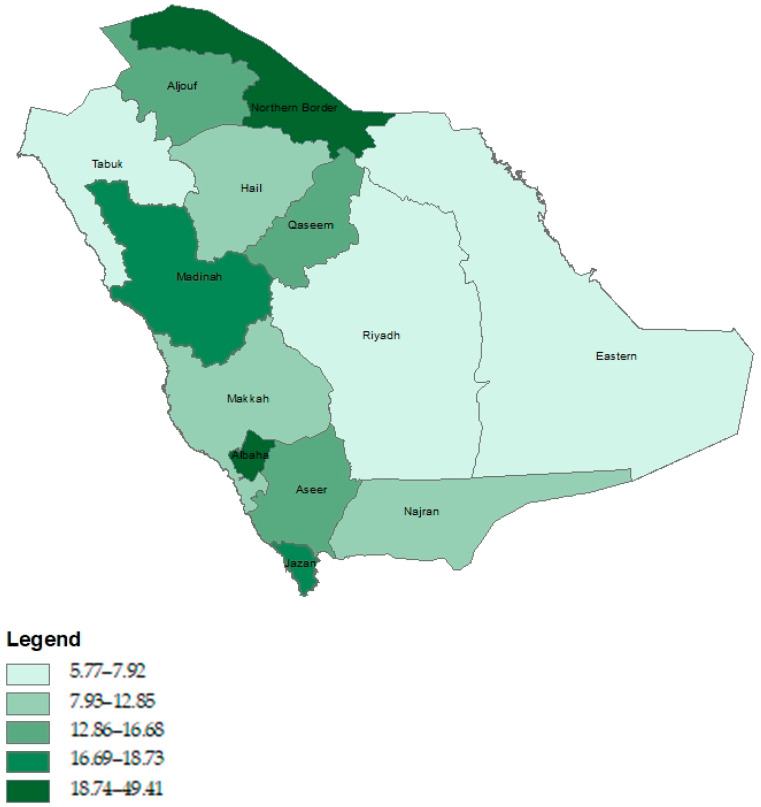
Epidemiological patterns of ICU e-referral requests across the Saudi administrative areas.

**Figure 4 healthcare-12-01939-f004:**
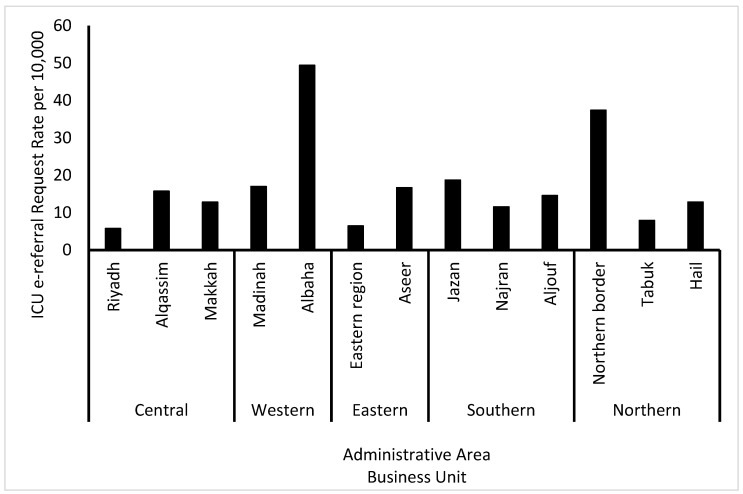
ICU e-referral rates per 10,000 population according to administrative area and business unit.

**Table 1 healthcare-12-01939-t001:** Sociodemographic characteristics of all patients with an ICU e-referral request.

Characteristics	Total N (%) N = 36,619
Age (µ, SD)	54.28 (20.46)
Sex	
Males	23,619 (64.49)
Females	13,000 (35.50)
Nationality	
Non-Saudi	10,419 (28.45)
Saudi	26,200 (71.55)
Year	
2020	16,921 (46.21)
2021	19,698 (53.79)

**Table 2 healthcare-12-01939-t002:** Sociodemographic characteristics of patients with an electronic ICU e-referral according to the five business units.

	Business Units N (%)				
Characteristics	Central 7067 (19.30)	Eastern 3308 (9.03)	Western 15,623 (42.66)	Northern 3927 (10.72)	Southern 6694 (18.28)
Age (µ, SD)	52.97 (21.05)	54.46 (20.02)	54.78 (19.40)	53.32 (20.82)	54.97 (22.11)
*p*-value	<0.001				
Sex					
Males	4580 (64.81)	1986 (60.03)	10,356 (66.29)	2365 (60.22)	4342 (64.86)
Females	2487 (35.19)	1322 (39.97)	5267 (33.71)	1562 (39.78)	2352 (35.14)
*p*-value	<0.001				
Nationality					
Non-Saudi	1829 (25.88)	698 (21.10)	5850 (37.44)	792 (20.17)	1250 (18.67)
Saudi	5238 (74.12)	2610 (78.90)	9773 (62.56)	3135 (79.83)	5444 (81.33)
*p*-value	<0.001				
Year					
2020	3125 (44.22)	1443 (43.62)	7685 (49.19)	1643 (41.84)	3025 (45.19)
2021	3942 (55.78)	1865 (56.38)	7938 (50.81)	2284 (58.16)	3669 (54.81)
*p*-value	<0.001				

**Table 3 healthcare-12-01939-t003:** E-referral characteristics and region of ICU e-referral requests in 2020 and 2021 across the Kingdom of Saudi Arabia.

Characteristics	Total 36,619 (100.00)	Central 7067 (19.30)	Eastern 3308 (9.03)	Western 15,623 (42.66)	Northern 3927 (10.72)	Southern 6694 (18.28)
Referral Types						
Out of scope	26,161 (71.44)	5979 (84.60)	2305 (69.68)	10,058 (64.38)	2971 (75.66)	4848 (72.42)
Overcapacity and bed shortage	7991 (21.82)	670 (09.48)	829 (25.06)	4701 (30.09)	571 (14.54)	1220 (18.22)
Unavailable equipment	2248 (6.14)	411 (5.82)	132 (3.99)	699 (4.47)	382 (9.73)	624 (9.32)
Compassionate request	219 (0.60)	7 (0.10)	42 (1.27)	165 (1.06)	3 (0.08)	2 (0.03)
*p*-value				<0.001		
Medical specialty						
Medical	22,878 (62.48)	3611 (51.10)	2218 (67.05)	10,546 (67.50)	2346 (59.74)	4157 (62.10)
Surgical	8104 (22.13)	1887 (26.70)	638 (19.29)	3182 (20.37)	924 (23.53)	1473 (22.00)
Cardiac surgery	4519 (12.34)	1330 (18.82)	346 (10.46)	1589 (10.17)	480 (12.22)	774 (11.56)
OB/GYN	795 (2.17)	169 (2.39)	63 (1.90)	210 (1.34)	110 (2.80)	243 (3.63)
Oncology	297 (0.81)	65 (0.92)	42 (1.27)	88 (0.56)	61 (1.55)	41 (0.61)
Organ transplant	26 (0.07)	5 (0.07)	1 (0.03)	8 (0.05)	6 (0.15)	6 (0.09)
*p*-value				<0.001		

**Table 4 healthcare-12-01939-t004:** Rate of e-referrals for ICU per 10,000 according to business units and administrative areas.

Business Units	Requests N (%) N = 36,619		Total Population N (%) N = 32,175,224	Ratio Request/Pop	Rate per 10,000
Central	7067 (19.30)		9,927,927 (30.85)	0.63	7.11
Western	15,623 (42.66)		10,498,620 (32.62)	1.31	14.88
Eastern	3308 (9.03)		5,125,254 (15.93)	0.57	6.45
Southern	6694 (18.28)		4,021,582 (12.50)	1.46	16.64
Northern	3927 (10.72)		2,601,841 (8.01)	1.34	15.10
**Business Units Regions**	**Administrative Area**	**Requests N (%)** **N = 36,619**	**Total Population N (%)** **N = 32,175,224**	**Ratio Request/Pop**	**Rate per 10,000**
Central	Riyadh	4961 (13.55)	8,591,748 (26.70)	0.51	5.77
	Alqassim	2106 (5.75)	1,336,179 (4.15)	1.39	15.76
Western	Makkah	10,308 (28.15)	8,021,463 (24.93)	1.13	12.85
	Madinah	3639 (9.94)	2,137,983 (6.64)	1.50	17.02
	Albaha	1676 (4.58)	339,174 (1.54)	2.97	49.41
Eastern	Eastern Region	3308 (9.03)	5,125,254 (15.92)	0.57	6.45
Southern	Aseer	3377 (9.22)	2,024,285 (6.29)	1.47	16.68
	Jazan	2632 (7.19)	1,404,997 (4.36)	1.65	18.73
	Najran	685 (1.87)	592,300 (1.84)	1.01	11.56
Northern	Aljouf	869 (2.37)	595,822 (1.85)	1.28	14.58
	Northern Border	1398 (3.82)	373,577 (1.16)	3.29	37.42
	Tabuk	702 (1.92)	886,036 (2.75)	0.70	7.92
	Hail	958 (2.62)	746,406 (2.31)	1.13	12.83

## Data Availability

The raw data supporting the conclusions of this article will be made available by the authors upon request.

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
