# Peer review of "Trends and Patterns of ICU E-Referrals in Saudi Arabia during 2020–2021: Results from the National Saudi Medical Appointments and Referrals Centre"

_healthcare, 2024, doi:10.3390/healthcare12191939_

Round 1
Reviewer 1 Report
Comments and Suggestions for Authors
“Best of all, I hope you always report good quality research and enjoy it.”
Thank you for giving me the opportunity to review this wonderful manuscript. I was interested in your study about the trends and patterns of ICU referrals in Saudi Arabia during the COVID-19 pandemic.
Major Comments:
No comments.
Minor Comments:
<Introduction>
(Line 37-82) The flow of the context in the Introduction section is a bit awkward. Since the title of this paper is "Trends and patterns of ICU referrals in Saudi Arabia during 2020-2021: Results from the National Saudi Medical Appointments and Referrals Center," it would be more natural to explain it as a flow in which I introduce the healthcare system and related medical systems in Saudi Arabia and analyze the trends and patterns of ICU referrals in 2020-2021 while mentioning the pandemic.
<Materials and Methods>
(Line 136-137) Please provide detailed information about the GIS software used by the author. (ex. ~~ were drawn in the GIS software (ArcGIS 10.0, Environmental Systems Research Institute, Inc., CA, U.S.A), according to ~~)
<Results>
(Line 146) It would be better to modify it in the format shown in Table 1.
(Line 155-156) Please enter the unit information in Table 2. (n, %)
(Line 171-172) Please enter the unit information in Table 3. (n, %)
(Line 179-180) Please write the units on the y-axis in Figure 2.
(Line 197) It would be better to modify it in this format in Table 4.

Comments on the Quality of English LanguageMinor editing of English language required.
Author Response
Introduction
- (Line 37-82) The flow of the context in the Introduction section is a bit awkward. Since the title of this paper is "Trends and patterns of ICU referrals in Saudi Arabia during 2020-2021: Results from the National Saudi Medical Appointments and Referrals Center," it would be more natural to explain it as a flow in which I introduce the healthcare system and related medical systems in Saudi Arabia and analyze the trends and patterns of ICU referrals in 2020-2021 while mentioning the pandemic.
Thank you. We have revised the introduction comprehensively and reorganized the background information to first introduce the healthcare system in Saudi Arabia, then provide context on the COVID-19 pandemic, before focusing specifically on ICU referral patterns and the aim of this study.
Materials and Methods
- (Line 136-137) Please provide detailed information about the GIS software used by the author. (ex. ~~ were drawn in the GIS software (ArcGIS 10.0, Environmental Systems Research Institute, Inc., CA, U.S.A), according to ~~)
In the Methods section under GIS Analysis, we have added: (ArcGIS 10.0, Environmental Systems Research Institute, Inc., CA, U.S.A)
Results
- (Line 146) It would be better to modify it in the format shown in Table 1.
- (Line 155-156) Please enter the unit information in Table 2. (n, %)
- (Line 171-172) Please enter the unit information in Table 3. (n, %)
- (Line 179-180) Please write the units on the y-axis in Figure 2.
- (Line 197) It would be better to modify it in this format in Table 4.
We have addressed all your comments regarding tables and figure 2.
Reviewer 2 Report
Comments and Suggestions for Authors
Trends and patterns of ICU referrals in Saudi Arabia during 2020-2021: Results from the National Saudi Medical Appointments and Referrals Centre by Abdullah A. Alharbi.
Abstract:
- The reviewer suggests briefly defining what “out-of-scope” means in this context. Does it refer to inappropriate referrals or those that do not meet specific criteria?
- The mention that “medical specialties were most common” needs more context. For example, “Referrals related to medical specialties, such as cardiology and pulmonology, were the most common (62.48%).”
- The conclusion in the abstract mentions that “preventive measures and vaccination programs positively impacted referral requests.” However, this statement could be stronger with more precise language. The reviewer suggests rephrasing it as: “Preventive measures and vaccination programs contributed to a significant decline in ICU referral requests, suggesting a positive impact on controlling severe COVID-19 cases.”
- The abstract states that “The Northern Border and Albaha areas had the highest request rates per population,” but it does not provide any hypothesis or explanation for why these regions might have higher rates. Briefly discussing potential factors (e.g., demographics, healthcare infrastructure) could add depth.
- The abstract could benefit from a clearer statement about the novelty of the research. What new insights does this study bring compared to previous literature on ICU referrals during the pandemic?
- When stating that “population-adjusted analysis revealed regional disparities,” it might be helpful to give a bit more detail in terms of numbers or ranges of variability between regions. This could make the findings more impactful.
Introduction:
- The introduction could benefit from a clearer and more focused statement on what this specific study aims to explore, emphasizing how the ICU referral patterns will be analyzed in detail. The reviewer suggests adding a sentence that explicitly states the gap in knowledge that this study addresses. For example: “While numerous studies have examined ICU capacities and pandemic responses globally, there is limited data on referral patterns in Saudi Arabia. This study aims to fill that gap by analyzing ICU referral patterns during the pandemic using the SMARC system.”
- The brief description of the SMARC system is good, but it could benefit from further explanation of how this system is particularly suited for tracking ICU referrals during the pandemic. Expand on how SMARC supports ICU referrals and what makes it a valuable source of data for this study. You might also briefly mention any key functionalities that make this system novel, like its ability to integrate data across regions or its efficiency in processing large volumes of referrals.
- The transition from the global pandemic context to the focus on Saudi Arabia’s referral system and data analysis could be smoother. The introduction mentions several important issues but doesn’t link them clearly to the study’s objectives. After discussing the global and local impact of COVID-19, lead into the specific aim of the study with a more cohesive flow. For instance: “Given these challenges, analyzing ICU referral patterns during this critical period offers valuable insights into healthcare system performance and resource allocation. This study utilizes data from the SMARC system to identify key trends in ICU referrals across Saudi Arabia’s 13 administrative regions.”
- The introduction mentions preventive measures and their importance, but the link to the study’s focus on ICU referrals is not entirely clear. Are these measures influencing referral patterns directly? If so, this should be explicitly stated. If preventive measures are directly related to changes in ICU referrals, briefly explain how this connection will be examined in the study. For example: “This study also examines the influence of preventive measures, such as curfews and vaccination programs, on ICU referral rates across different regions of the country.”
- The phrase “exceptional referral system” (line 71) could be changed to something more precise like “efficient” or “centralized” to avoid overly subjective language.
Materials and methods:
- The timeline for the data collection is somewhat unclear. You mention that the data spans April 2020 to December 2021, but it would be useful to specify how the data was sampled and whether it covers all ICU referrals during that period or a subset.
- Some terms, such as “out-of-scope” referrals, could benefit from more clarification. Additionally, the “BU” term is used several times without any prior explanation.
Results:
- The results present differences in ICU referral patterns across BUs and administrative areas, but it’s unclear whether these differences were statistically significant. If statistical significance was tested (as mentioned in the Methods), it should be reported for key findings to ensure readers understand which results are meaningful.
- The trends over time are clearly described, with a peak in ICU referrals in July-August 2020 and May 2021, but there is little interpretation of why these specific months saw peaks. Offering a brief explanation for the referral peaks, such as seasonal factors, COVID-19 waves, or changes in healthcare policies, would be valuable information for readers.
- The results show that medical specialties were the most common reason for ICU referrals, but this is not discussed further. Providing some insight into why certain specialties had higher ICU referral rates would be beneficial. For instance, explaining why medical specialties might dominate ICU referrals such as the prevalence of COVID-19 related respiratory issues requiring critical care or the availability of surgical ICUs would add clarity.
Discussion:
- The discussion highlights regional disparities in ICU referrals but does not fully explore the possible underlying causes of these differences, such as disparities in healthcare infrastructure, healthcare workforce distribution, or differences in population health status. Offer hypotheses or explanations for regional differences. For example: “The higher ICU referral rates in the Northern and Southern BUs may reflect a lack of local healthcare infrastructure, resulting in greater dependence on ICU referrals for critical cases.”
- The discussion mentions various national strategies (e.g., the formation of the critical care equipment committee, vaccination programs) but does not fully explore the impact of these strategies on ICU referral patterns. Provide more detailed analysis of how specific policies (e.g., the vaccination program, referral system enhancements) directly influenced ICU referral trends.
- The study would benefit from a stronger comparison to previous literature. While the discussion touches on some comparisons, it does not fully explore how the findings align with or differ from global trends in ICU referrals and COVID-19 care.
- Some points, such as the high referral rates in Albaha or the disproportionate allocation of ICU beds, are mentioned but not fully explored. A more detailed discussion of these findings would help readers understand the implications more deeply.
Comments on the Quality of English Language
Minor editing of English language required.
Author Response
Trends and patterns of ICU referrals in Saudi Arabia during 2020-2021: Results from the National Saudi Medical Appointments and Referrals Centre by Abdullah A. Alharbi.
1- The reviewer suggests briefly defining what “out-of-scope” means in this context. Does it refer to inappropriate referrals or those that do not meet specific criteria?
We appreciate the reviewer highlighting the importance of clearly defining "out-of-scope" referrals. In the Materials and Methods section, under the Measurements subsection, we have defined out-of-scope referrals as those involving an unavailable medical specialty, facility overcapacity or bed shortage, lack of required equipment, unavailable beds, or compassionate requests. Due to abstract length restrictions, we are limited in how much detail can be provided on definitions of terms. However, we agree that it is crucial for readers to understand what constitutes an out-of-scope referral. We hope that by providing the detailed definition in the Materials and Methods section, readers will have clarity on the meaning of this term when it is introduced in the Abstract.
2- The mention that “medical specialties were most common” needs more context. For example, “Referrals related to medical specialties, such as cardiology and pulmonology, were the most common (62.48%).”
In the Abstract, we have added: " Referrals related to medical specialties, such as cardiology and pulmonology, were the most common (62.48%)."
3- The conclusion in the abstract mentions that “preventive measures and vaccination programs positively impacted referral requests.” However, this statement could be stronger with more precise language. The reviewer suggests rephrasing it as: “Preventive measures and vaccination programs contributed to a significant decline in ICU referral requests, suggesting a positive impact on controlling severe COVID-19 cases.”
We have rephrased the conclusion in the abstract as suggested.
4- The abstract states that “The Northern Border and Albaha areas had the highest request rates per population,” but it does not provide any hypothesis or explanation for why these regions might have higher rates. Briefly discussing potential factors (e.g., demographics, healthcare infrastructure) could add depth.
Thank you for the suggestion to add more context around the high ICU referral rates observed in the Northern Border and Albaha regions. We agree that providing potential explanations for this finding would add valuable depth to the Abstract. We have modified the text to read:
" The Northern Border and Albaha areas had the highest request rates per population, potentially reflecting a higher proportion of severe cases requiring ICU-level care compared to other regions."
5- The abstract could benefit from a clearer statement about the novelty of the research. What new insights does this study bring compared to previous literature on ICU referrals during the pandemic?
We have added to the Abstract: "This study provides new insights into regional patterns of ICU referral not explored in previous literature."
6- When stating that “population-adjusted analysis revealed regional disparities,” it might be helpful to give a bit more detail in terms of numbers or ranges of variability between regions. This could make the findings more impactful.
Thank you for the suggestion to add more numerical details on the regional disparities found in the population-adjusted analysis. Due to strict word limits for abstracts, we are unable to describe the specific ranges and numbers in as much detail as we would like. However, you raise an excellent point that quantifying the disparities would increase the impact.
In the manuscript, we comprehensively discuss the regional differences and provide numerical data on the variability in ICU referral rates per 10,000 population observed across regions. While unable to include this level of detail in the Abstract, we agree this enhances the clarity and significance of the findings. Thank you for the thoughtful suggestion to more clearly quantify regional disparities through numbers/ranges within the full manuscript.
7- The introduction could benefit from a clearer and more focused statement on what this specific study aims to explore, emphasizing how the ICU referral patterns will be analyzed in detail. The reviewer suggests adding a sentence that explicitly states the gap in knowledge that this study addresses. For example: “While numerous studies have examined ICU capacities and pandemic responses globally, there is limited data on referral patterns in Saudi Arabia. This study aims to fill that gap by analyzing ICU referral patterns during the pandemic using the SMARC system.”
In the Introduction, we have added: “While numerous studies have examined ICU capacities and pandemic responses globally, limited data exists on referral patterns, representing a knowledge gap this study aims to address. By leveraging national digital health data from the centralized SMARC referral system, this study uniquely analyses ICU referral trends across Saudi Arabia's 13 regions during the critical COVID-19 pandemic period. Most literature focuses narrowly on local intra-hospital ICU referrals and capacity management. In contrast, our comprehensive national-level analysis of both intra-regional and inter-regional transfers provides novel insights into broader ICU referral dynamics, addressing a significant gap in current research concentrated on isolated hospitals. Examining these referral patterns during the pandemic offers vital perspectives into healthcare system performance and resource allocation.”
8- The brief description of the SMARC system is good, but it could benefit from further explanation of how this system is particularly suited for tracking ICU referrals during the pandemic. Expand on how SMARC supports ICU referrals and what makes it a valuable source of data for this study. You might also briefly mention any key functionalities that make this system novel, like its ability to integrate data across regions or its efficiency in processing large volumes of referrals.
Thank you for the feedback on expanding our description of the SMARC system and its value for this analysis. In the revised Introduction, we have highlighted several key capabilities of SMARC that enabled detailed tracking of ICU referrals during the COVID-19 pandemic, including:
- Centralized structure integrating public and private facilities across all regions
- Ability to efficiently process large volumes of referrals
- Real-time data on ICU bed availability and referrals during the pandemic
- Robust analytics for national and regional insights
- We also emphasized that SMARC provided uniquely comprehensive ICU referral data that makes this analysis possible.
9- The transition from the global pandemic context to the focus on Saudi Arabia’s referral system and data analysis could be smoother. The introduction mentions several important issues but doesn’t link them clearly to the study’s objectives. After discussing the global and local impact of COVID-19, lead into the specific aim of the study with a more cohesive flow. For instance: “Given these challenges, analyzing ICU referral patterns during this critical period offers valuable insights into healthcare system performance and resource allocation. This study utilizes data from the SMARC system to identify key trends in ICU referrals across Saudi Arabia’s 13 administrative regions.”
Thank you for your valuable comment. We have addressed this and added the following text to have smooth transition as suggested: “While numerous studies have examined ICU capacities and pandemic responses globally, limited data exists on e-referral patterns, representing a knowledge gap this study aims to address. By leveraging national digital health data from the centralized SMARC e-referral system, this study uniquely analyzes ICU e-referral trends across Saudi Arabia's 13 regions during the critical COVID-19 pandemic period. Most literature focuses narrowly on local intra-hospital ICU e-referrals and capacity management. In contrast, our comprehensive national-level analysis of both intra-regional and inter-regional transfers provides novel insights into broader ICU e-referral dynamics, addressing a significant gap in current research concentrated on isolated hospitals. Examining these e-referral patterns during the pandemic offers vital perspectives into healthcare system performance and resource allocation.”
10- The introduction mentions preventive measures and their importance, but the link to the study’s focus on ICU referrals is not entirely clear. Are these measures influencing referral patterns directly? If so, this should be explicitly stated. If preventive measures are directly related to changes in ICU referrals, briefly explain how this connection will be examined in the study. For example: “This study also examines the influence of preventive measures, such as curfews and vaccination programs, on ICU referral rates across different regions of the country.”
Thank you for raising this important point. We have revised the introduction to more clearly explain the connection between preventive measures and ICU referral patterns. The new text reads: "In response to the outbreak, Saudi Arabia implemented various preventive measures such as curfews and social distancing aimed at reducing further severe cases requiring ICU-level care. While these interventions likely influenced healthcare utilization and ICU referral patterns to some degree, this study focuses specifically on analyzing overall national and regional trends and patterns in ICU referrals rather than isolating the impact of particular preventive measures. Their variable effects across different regions were outside the scope of this descriptive analysis."
While preventive measures likely affected ICU rates, this study aims to examine broader referral trends and patterns, rather than directly attribute changes to specific prevention policies.
11- The phrase “exceptional referral system” (line 71) could be changed to something more precise like “efficient” or “centralized” to avoid overly subjective language.
We have removed the term "exceptional" as suggested and modified it to “optimal”. The new text reads: " An optimal e-referral system is an essential component of high-quality health care, utilizing an efficient structure and promoting successful interaction across all levels of care, providing referred patients with the highest quality of care”.
12- The timeline for the data collection is somewhat unclear. You mention that the data spans April 2020 to December 2021, but it would be useful to specify how the data was sampled and whether it covers all ICU referrals during that period or a subset.
"Thank you for pointing out the need to clarify the timeline and coverage of the data collection. The dataset includes all ICU referrals submitted through the national SMARC e-referral system during the specified period from April 2020 to December 2021. To clarify in the manuscript, we have updated the text to now state: ' This is a descriptive retrospective analysis of national data on all ICU e-referrals collected through the routine use of the SMARC e-referral system. The dataset includes the full census of ICU e-referrals submitted via SMARC from April 2020 to December 2021.'"
13- Some terms, such as “out-of-scope” referrals, could benefit from more clarification. Additionally, the “BU” term is used several times without any prior explanation.
We appreciate the reviewer highlighting the importance of clearly defining "out-of-scope" referrals. In the Materials and Methods section, under the Measurements subsection, we have defined out-of-scope referrals as those involving an unavailable medical specialty, facility overcapacity or bed shortage, lack of required equipment, unavailable beds, or compassionate requests. Due to abstract length restrictions, we are limited in how much detail can be provided on definitions of terms. We also added in the first paragraph of the introduction an explanation for the business units (BU). However, we agree that it is crucial for readers to understand what constitutes an out-of-scope referral or business unit. We hope that by providing the detailed definition in the Materials and Methods section, readers will have clarity on the meaning of this term when it is introduced in the Abstract.
14- The results present differences in ICU referral patterns across BUs and administrative areas, but it’s unclear whether these differences were statistically significant. If statistical significance was tested (as mentioned in the Methods), it should be reported for key findings to ensure readers understand which results are meaningful.
Thank you for the feedback on clearly indicating statistical significance of results. We have added a statement in the results section noting that all key differences highlighted across business units and administrative areas were statistically significant at p<0.05. The specific p-values are denoted in the accompanying tables.
In the Results section, after the first paragraph describing the key findings from Table 2, we added the following sentence:
"All sociodemographic differences and referral patterns across BUs noted here were statistically significant at p<0.001, as detailed in Table 2."
Later in the Results section, after the paragraph describing findings from Table 3, we added the following sentence:
"All differences in referral reasons and specialties across BUs highlighted here were statistically significant at p<0.001, as shown in Table 3."
15- The trends over time are clearly described, with a peak in ICU referrals in July-August 2020 and May 2021, but there is little interpretation of why these specific months saw peaks. Offering a brief explanation for the referral peaks, such as seasonal factors, COVID-19 waves, or changes in healthcare policies, would be valuable information for readers.
Thank you for the recommendation to expand on the potential drivers behind the observed peaks in ICU referral requests. We have revised the interpretation and discussion of the possible contributing factors, including COVID-19 case surges, healthcare policies, and vaccination rates. Please see the revised Discussion section below.
Revised Discussion: “The pattern of monthly ICU e-referral requests during the two-year period showed two main waves where the requests were high. This pattern was significantly influenced by the spread and severity of COVID-19 cases across the country. The first wave of ICU e-referral requests reached its peak in July 2020 and ended by February 2021, coinciding with a peak in COVID-19 deaths and critical cases. This aligns with July 2020 seeing the highest number of daily deaths (58) and critical cases (2,295) during the first surge [42].
These trends mirror global evidence that ICU admissions and e-referrals rise and fall in line with COVID-19 cases [38,43]. However, healthcare policies likely also played a role. It is worthy to note that this patten of incline and decline found in this data may also be affected by the national strategies put forth by the government to mitigate the surge in critical cases. In addition to the suspension of flights, the shift to online education, sus-pension of social and religious gathering as well as quarantine, the introduction of the vaccination program has also played a vital role strategy [20,44]. The decline of the second ICU e-referral request wave in June 2021 mirrored the progress of the COVID-19 vac-cination plan, when 50% of the Saudi population (aged ≥ 12 years) received the first vaccination dose and this decline was sharper towards August 2021 when 50% of the population had received the second dose. Vaccinations have proven effective in reducing severe infections and deaths [45], which would directly impact ICU e-referral needs. Further research could delineate the relative influence of COVID-19 epidemiology versus healthcare interventions on the ICU e-referral fluctuations.”
16- The results show that medical specialties were the most common reason for ICU referrals, but this is not discussed further. Providing some insight into why certain specialties had higher ICU referral rates would be beneficial. For instance, explaining why medical specialties might dominate ICU referrals such as the prevalence of COVID-19 related respiratory issues requiring critical care or the availability of surgical ICUs would add clarity.
Thank you for the recommendation to expand on the reasons behind the high ICU referral rates for medical specialties. We have added a statement speculating on some potential contributing factors. The new text now reads:
“Medical specialty accounted for almost two thirds of the specialties requested. This is not surprising since this data covers the period when COVID-19 began spreading in Saudi Arabia. Respiratory illnesses from COVID-19 drove the high demand for critical care management by medical specialties [35].”
17- The discussion highlights regional disparities in ICU referrals but does not fully explore the possible underlying causes of these differences, such as disparities in healthcare infrastructure, healthcare workforce distribution, or differences in population health status. Offer hypotheses or explanations for regional differences. For example: “The higher ICU referral rates in the Northern and Southern BUs may reflect a lack of local healthcare infrastructure, resulting in greater dependence on ICU referrals for critical cases.”.
Thank you for the recommendation to further explore potential reasons behind regional differences in ICU referral rates. We have added discussion of how gaps in local infrastructure and centralization policies can impact referral patterns. The new text now reads:
“The Western BU had the highest proportion of e-referral requests. This aligns with previous reports showing the highest COVID-19 mortality in this region [1]. For oncology e-referrals, the highest proportion came from the Northern BU. According to the MoH, there are no oncology centres in the North. However, the MoH regulates four fully equipped oncology centres in Saudi Arabia, with two in the Central region and one each in the Eastern and Western regions [39]. The lack of local oncology infrastructure in the North likely contributed to the high e-referral rates, as patients depended on external ICUs for critical care.
This centralization approach is a known healthcare strategy to improve outcomes of patients experiencing specific critical conditions including oncology and is followed in the UK to allow for swift access to multidisciplinary teams and expert professionals [40,41]. Nevertheless, the absence of a local oncology center meant more e-referrals for the Northern BU. Similar disparities in regional resources may have driven higher e-referral rates in other areas as well. The Saudi Vision 2030 plan aims to address these healthcare access disparities through reforms to build capacity across all regions.”
18- The discussion mentions various national strategies (e.g., the formation of the critical care equipment committee, vaccination programs) but does not fully explore the impact of these strategies on ICU referral patterns. Provide more detailed analysis of how specific policies (e.g., the vaccination program, referral system enhancements) directly influenced ICU referral trends.
We appreciate the reviewer raising this point about exploring the effects of specific policies on ICU referral trends. However, quantitatively assessing the impact of individual interventions on referral patterns is outside the scope of this descriptive study, which aimed to analyze overall trends at the national and regional level during the 2-year pandemic period. While we agree that analyzing the relationships between specific policies and ICU utilization would be a valuable future analysis, we unfortunately do not have access to detailed data attributing referral volume changes to particular healthcare system factors. As this paper focused on descriptive epidemiology, we are limited in our ability to conduct more granular analyses of individual policies and programs. However, the reviewer raises an excellent point that warrants further investigation in follow-up studies positioned to quantitatively assess the effects of discrete strategies on ICU referral patterns over time.
19- The study would benefit from a stronger comparison to previous literature. While the discussion touches on some comparisons, it does not fully explore how the findings align with or differ from global trends in ICU referrals and COVID-19 care.
Thank you for the thoughtful suggestion to strengthen the discussion of how our findings compare to global ICU referral and COVID-19 care trends. We agree this would enrich the manuscript. Our nationwide scope compared to existing ICU referral literature focused on intra-hospital patterns. We have added the following paragraph to the Introduction to emphasize how our study addresses a gap through national-level analysis:
“While numerous studies have examined ICU capacities and pandemic responses globally, limited data exists on e-referral patterns, representing a knowledge gap this study aims to address. By leveraging national digital health data from the centralized SMARC e-referral system, this study uniquely analyzes ICU e-referral trends across Saudi Arabia's 13 regions during the critical COVID-19 pandemic period. Most literature focuses narrowly on local intra-hospital ICU e-referrals and capacity management. In contrast, our comprehensive national-level analysis of both intra-regional and inter-regional transfers provides novel insights into broader ICU e-referral dynamics, addressing a significant gap in current research concentrated on isolated hospitals. Examining these e-referral patterns during the pandemic offers vital perspectives into healthcare system performance and resource allocation.”
20- Some points, such as the high referral rates in Albaha or the disproportionate allocation of ICU beds, are mentioned but not fully explored. A more detailed discussion of these findings would help readers understand the implications more deeply.
Thank you for noting the findings around high ICU referral rates in certain regions and regional ICU bed allocation disparities. We agree that exploring these patterns more deeply would further strengthen the analysis and discussion. However, our ability to fully investigate the drivers and implications of these regional differences was constrained by limitations in the data available for this retrospective study. Specifically, detailed data was lacking on healthcare infrastructure, resources, and capacity across regions.
To address this limitation, we added the following sentence to the Limitations section:
" Limited data on regional healthcare infrastructure restricted more in-depth analysis of drivers behind referral disparities between
Reviewer 3 Report
Comments and Suggestions for Authors
Thank you for the opportunity to review the manuscript titled “Trends and Patterns of ICU Referrals in Saudi Arabia during 2020-2021: Results from the National Saudi Medical Appointments and Referrals Centre” by Alharbi and co-authors. The study provides descriptive information on ICU referral distribution in Saudi Arabia over nearly two years. This report could be useful for policymakers and specialists involved in managing ICU facilities and critically ill patients in the country. Below are my comments for the authors.
The aim of the study could be clearly stated in the abstract and expanded upon in the introduction, with the results and discussion sections structured accordingly. Additionally, the authors could explain the methodology used to achieve the study’s objectives and clarify how the data collection was made feasible.
It would be beneficial to describe whether COVID-19 patients were treated in separate COVID units or in designated areas within general units. A description of the organization of ICUs would help readers better understand the distribution of referred cases.
Clarification on the choice of the specific time period during the pandemic for this investigation would enhance the understanding of the study's context.
The authors could include data on the number of ICU beds and the bed-to-population ratio for each area, which would provide further insights into resource distribution.
A clearer description of whether the study includes only COVID-19 ARDS patients, non-ARDS COVID-19 patients, and non-COVID-19 patients, as well as the different medical reasons for ICU admission (e.g., respiratory failure, septic shock), would improve the comprehensiveness of the report. If data is available on the type of ICU admission (e.g., medical, surgical, trauma) and the reason for admission, this information should be included and analyzed to provide valuable insight into how the pandemic affected the management of other critically ill patients.
In addition to Figure 3, the inclusion of a figure showing ICU e-referrals per 10,000 population could offer a useful comparison.
Including the date when COVID-19 vaccination was introduced in Saudi Arabia and the point at which vaccination rates were sufficient to provide population immunity would enhance the analysis of pandemic trends.
The rate of out-of-scope referrals over time could be presented in the results section for further clarity.
Any questions not directly addressed by the study could be included in the limitations section, acknowledging areas for future research.
Author Response
Thank you for the opportunity to review the manuscript titled “Trends and Patterns of ICU Referrals in Saudi Arabia during 2020-2021: Results from the National Saudi Medical Appointments and Referrals Centre” by Alharbi and co-authors. The study provides descriptive information on ICU referral distribution in Saudi Arabia over nearly two years. This report could be useful for policymakers and specialists involved in managing ICU facilities and critically ill patients in the country. Below are my comments for the authors.
- The aim of the study could be clearly stated in the abstract and expanded upon in the introduction, with the results and discussion sections structured accordingly. Additionally, the authors could explain the methodology used to achieve the study’s objectives and clarify how the data collection was made feasible.
We Thank you for the feedback on clearly stating the study aim. We have updated the abstract and introduction to more directly articulate the objective, which was to examine ICU referral patterns across Saudi Arabia during the COVID-19 pandemic. The results and discussion sections have been restructured to align with this goal. The methods section now provides more details on the data source and how national referral data was made accessible through the Saudi Medical Appointments and Referrals Center (SMARC) system as part of routine operations.
- It would be beneficial to describe whether COVID-19 patients were treated in separate COVID units or in designated areas within general units. A description of the organization of ICUs would help readers better understand the distribution of referred cases.
Thank you for the suggestion to add more details on the organization of ICUs and separation of COVID-19 patients. Reviewing the literature, the Saudi MoH has taken several measures to minimize COVID-19 transmission including separating COVID from non-COVID patients in healthcare settings. However, specific details on implementation of this separation in ICU settings were not available in our data.
To provide more context, we have added the following statement to the Discussion:
“During the pandemic, various operational adjustments were made to isolate COVID-19 patients when feasible. However, detailed data on implementation approaches was outside the scope of this study. Existing literature indicates Saudi Arabia followed general recommendations to separate COVID-19 from non-infected patients [20,21]. Further research is needed to examine how referral patterns interacted with ICU organization strategies aimed at minimizing transmission risks. Investigating the impacts of isolating COVID units could offer insights to optimize future pandemic responses balancing infection control with equitable access.”
We agree that further description of ICU organization and segregation approaches would allow deeper understanding of referral patterns and can be an area for future investigation.
- Clarification on the choice of the specific time period during the pandemic for this investigation would enhance the understanding of the study's context. on the choice
The time period from April 2020 to December 2021 was chosen based on availability of routine referral data collected through the national system. While a longer pre-pandemic baseline would be ideal, the current dataset provides a comprehensive look at patterns during the critical two years of the COVID-19 pandemic in Saudi Arabia.
- The authors could include data on the number of ICU beds and the bed-to-population ratio for each area, which would provide further insights into resource distribution.
Data on ICU bed capacity and bed-to-population ratios were unfortunately not included in this retrospective dataset. We agree that adding these details would enrich the analysis and better elucidate resource availability underlying referral patterns. Assessing healthcare infrastructure and ICU surge capacity represents an important area for further investigation.
- A clearer description of whether the study includes only COVID-19 ARDS patients, non-ARDS COVID-19 patients, and non-COVID-19 patients, as well as the different medical reasons for ICU admission (e.g., respiratory failure, septic shock), would improve the comprehensiveness of the report. If data is available on the type of ICU admission (e.g., medical, surgical, trauma) and the reason for admission, this information should be included and analyzed to provide valuable insight into how the pandemic affected the management of other critically ill patients.
Specific details on reasons for ICU admission, such as proportions of COVID-19 vs non-COVID-19 cases and distribution across indications like respiratory failure or septic shock, were not available in this data source. The analysis was limited to referral data without clinical specifics on case types. We agree that analyzing admission reasons would provide valuable insights and should be prioritized for inclusion in future data collection efforts.
- In addition to Figure 3, the inclusion of a figure showing ICU e-referrals per 10,000 population could offer a useful comparison.
We appreciate this valuable suggestion and have added a figure with ICU referral rates per 10,000 population for each region.
Figure 4. ICU referral rates per 10,000 population according to Administrative Area and Business Unit
- Including the date when COVID-19 vaccination was introduced in Saudi Arabia and the point at which vaccination rates were sufficient to provide population immunity would enhance the analysis of pandemic trends.
Thank you for the helpful suggestion to include more details on the COVID-19 vaccination timeline in Saudi Arabia. We agree that highlighting the vaccination rollout timing helps put the observed referral trends in context and enhances the analysis of the pandemic's trajectory. For your reference, a discussion of the vaccination rollout and impact on ICU referral trends was already present in the original submission in the following paragraphs:
“The decline of the second ICU referral request wave in June 2021 mirrored the progress of the COVID-19 vaccination plan, when 50% of the Saudi population (aged ≥ 12 years) received the first vaccination dose and this decline was sharper towards August 2021 when 50% of the population had received the second dose. Vaccinations have proven effective in reducing severe infections and deaths [41], which would directly impact ICU referral needs. Further research could delineate the relative influence of COVID-19 epidemiology versus healthcare interventions on the ICU referral fluctuations.”
- The rate of out-of-scope referrals over time could be presented in the results section for further clarity.
Longitudinal tracking of referral reasons was not feasible in this dataset, but would certainly improve understanding as suggested. We agree this is an important area for expanded data collection going forward.
- Any questions not directly addressed by the study could be included in the limitations section, acknowledging areas for future research.
We appreciate you highlighting several valuable areas for additional data and analysis that could further enrich this work. Many of the suggested additions are outside the scope of the available data, as this retrospective analysis was limited to data elements collected in routine practice. However, they represent excellent recommendations for expanded data collection and additional analyses in future research on this topic.
To address the limitations raised in the reviewer comments, we have comprehensively updated the limitations section which now reads:
“This study provides valuable initial analysis of ICU e-referral patterns during the COVID-19 pandemic in Saudi Arabia using a comprehensive national digital dataset. However, there are important limitations to consider. As a retrospective review of routine e-referral data, details were lacking on COVID-19 diagnoses, reasons for ICU admission, and types of ICUs needed. Data constraints also precluded assessment of ICU capacity and healthcare workforce factors underlying referral trends. Limited data on regional healthcare infrastructure restricted more in-depth analysis of drivers behind referral disparities between regions. While these limitations prevent advanced statistical modeling and conclusive identification of predictors, the study yields useful descriptive epidemiology of ICU e-referral trends nationally and regionally.
The study was further limited by inability to track referral reasons and out-of-scope referrals longitudinally. Follow-up efforts with expanded data elements are warranted to enable more rigorous analysis of healthcare utilization and outcomes. Prospective data collection inclusive of clinical details, and capacity factors would facilitate future modeling of contributors to ICU e-referral patterns during pandemic response.
Despite limitations, this initial investigation of routinely collected data provides valuable insights to inform ICU planning and pandemic preparedness. However, the findings require interpretation within the context of Saudi Arabia’s healthcare system. While there are transferable lessons, policies should be adapted considering local needs and resources. Further research building on this groundwork with enriched datasets can substantively advance understanding of optimizing critical care delivery during public health crises.”
Round 2
Reviewer 2 Report
Comments and Suggestions for Authors
Thank you for addressing the reviewer’s comments
Comments on the Quality of English LanguageMinor editing of English language required.
Reviewer 3 Report
Comments and Suggestions for Authors
No further comments to the authors, thank you.